# The Ten-Year Risk Prediction for Cardiovascular Disease for Malaysian Adults Using the Laboratory-Based and Office-Based (Globorisk) Prediction Model

**DOI:** 10.3390/medicina58050656

**Published:** 2022-05-12

**Authors:** Che Muhammad Nur Hidayat Che Nawi, Mohd Azahadi Omar, Thomas Keegan, Yong-Poh Yu, Kamarul Imran Musa

**Affiliations:** 1Department of Community Medicine, School of Medical Sciences, Universiti Sains Malaysia, Kubang Kerian 16150, Kelantan, Malaysia; drchehidayat@student.usm.my; 2Sector for Biostatistics & Data Repository, National Institutes of Health, Ministry of Health, Shah Alam 40170, Selangor, Malaysia; drazahadi@moh.gov.my; 3Faculty of Health and Medicine, Lancaster Medical School, Lancaster University, Lancaster LA1 4YW, UK; t.keegan@lancaster.ac.uk; 4Redbeat Academy, AirAsia Berhad, Kuala Lumpur 50470, Selangor, Malaysia; richieyyp@gmail.com

**Keywords:** cardiovascular disease, ten-year risk prediction, Globorisk, Malaysia

## Abstract

*Background and Objectives*: Globorisk is a well-validated risk prediction model that predicts cardiovascular disease (CVD) in the national population of all countries. We aim to apply the Globorisk calculator and provide the overall, sex-specific, ethnic-specific, region-specific, and state-specific 10-year risk for CVD among Malaysian adults. *Materials and Methods:* Using Malaysia’s risk factor levels and CVD event rates, we calculated the laboratory-based and office-based risk scores to predict the 10-year risk for fatal CVD and fatal plus non-fatal CVD for the Malaysian adult population. We analysed data from 8253 participants from the 2015 nationwide Malaysian National Health and Morbidity Survey (NHMS 2015). The average risk for the 10-year fatal and fatal plus non-fatal CVD was calculated, and participants were further grouped into four categories: low risk (<10% risk for CVD), high risk A (≥10%), high risk B (≥20%), and high risk C (≥30%). *Results:* Results were reported for all participants and were then stratified by sex, ethnicity, region, and state. The average risks for laboratory-based fatal CVD, laboratory-based fatal plus non-fatal CVD, and office-based fatal plus non-fatal CVD were 0.07 (SD = 0.10), 0.14 (SD = 0.12), and 0.11 (SD = 0.09), respectively. *Conclusions:* There were substantial differences in terms of the sex-, ethnicity- and state-specific Globorisk risk scores obtained.

## 1. Introduction

Cardiovascular diseases (CVDs) are the leading cause of death worldwide, accounting for over 17 million deaths in 2016, 31% of the world total. The goal set by the World Health Organization (WHO) is to reduce CVD-related premature death by 30% [1]. CVD fatalities accounted for 21.7% of all hospital deaths in Malaysia in 2017 [2].

CVD-related deaths could be reduced by predicting the CVD risk and, subsequently, mitigating CVD risk factors. To determine the number of people at high risk for CVD, a good model of CVD risk prediction and a nationally representative cardiometabolic profile are required (e.g., those with a CVD risk greater than 30 percent). The measurement of CVD risk will aid in tracking progress toward global non-communicable disease (NCD) treatment goals.

Framingham, INTERHEART, SCORE, WHO/ISH CVD, and Pooled Cohort Equation are a few of the current CVD risk calculators that have been tested on populations [3,4,5,6,7,8]. However, risk predictions developed for a specific population cannot be used in other populations or the same populations later because the mean levels of CVD predictors vary across populations and over time [9,10,11,12,13,14]. Two examples of the recommended risk prediction models that are tailored for specific populations are the SCORE model, which is tailored for European countries using national mortality statistics, and Globorisk [12,15,16]. Unlike the SCORE model, Globorisk provides country-specific CVD risk scores to estimate the 10-year risk of fatal and non-fatal CVD. Its model has been validated and calibrated using data from 182 countries [15,16,17].

Globorisk is a cardiovascular disease risk score that has good internal and external validation to predict the risk of heart attack or stroke in healthy people (those who have not experienced a heart attack or stroke) because, during the recalibration procedure, the mean risk factor levels and the CVD event rate were based on the best estimate for the target country [16,17]. It predicts the likelihood of a heart attack or stroke in the following ten years based on a person’s country of residence, age, sex, smoking status, diabetes, systolic blood pressure, and total cholesterol (laboratory-based risk prediction). For someone who is not sure about their diabetes status or cholesterol level, they can use Globorisk’s office-based risk prediction available at this link: http://www.globorisk.org/ (accessed on 15 October 2019), which requires information about age, sex, systolic blood pressure, body mass index (BMI), and smoking status.

There has been no local study that has used Globorisk to calculate the CVD risk among the Malaysian population. However, knowing the average risk for fatal and fatal plus non-fatal CVD and the proportion of the Malaysian population with specific risk factors for developing a CVD event in the next 10 years will help public health workers, epidemiologists, and policymakers in Malaysia to guide CVD control and prevention programmes.

The study aimed to apply the Globorisk calculator and provide the overall, sex-specific, ethnic-specific, region-specific, and state-specific 10-year risk for CVD among Malaysian adults. Three measures of CVD risk can be calculated using the Globorisk risk prediction model: (1) the laboratory-based 10-year risk of fatal CVD, (2) the laboratory-based 10-year risk of fatal plus non-fatal CVD, and (3) the office-based 10-year risk of fatal plus non-non-fatal CVD.

## 2. Materials and Methods

### 2.1. Survey Data Source and Variables

The Institute of Public Health, Ministry of Health, Malaysia, provided the National Health and Morbidity Survey (NHMS 2015) data based on their nationwide population-based cross-sectional survey in 2015. It is a dataset of nationally representative cardiometabolic risk factor profiles of Malaysians. To obtain permission to reanalyse the NHMS 2015 data, we obtained ethical approval from the Medical Research and Ethics Committee, Ministry of Health Malaysia (see Ethical Approval section). Other studies have published their results using the same dataset as ours (the NHMS 2015 dataset) [18,19].

The information on the NHMS 2015, including the methodology used, is available from the Malaysia Ministry of Health webpage at this link: https://www.moh.gov.my/moh/resources/nhmsreport2015vol2.pdf (accessed on 10 April 2019). Briefly, diabetes status was characterised as fasting blood glucose (finger-prick sample) tested using the CardioChek portable blood test device with a value of more than 7.1 mmol/L or the recorded use of oral hypoglycaemic agents or insulin injection. The use of finger-prick sample (capillary origin) is optimal when used for point-of-care non-clinical setting and along with venous plasma glucose can be used for diagnosis of diabetes in general practice [20,21].

Hypertension was defined as a blood pressure measurement using Omron’s digital automated blood pressure monitor model HEM-907 with a value of systolic blood pressure and/or diastolic blood pressure greater than or equal to 140 mm Hg and/or 90 mm Hg, respectively. Meanwhile, hypercholesterolemia was defined as a total cholesterol level (finger-prick sample) tested using the portable CardioChek blood test device. Smoking status was defined as whether the participant was currently smoking.

### 2.2. Selection of Participants

The NMHS 2015 dataset contains data on 29,460 participants (14,225 men and 15,235 women). Participants were eligible if older than 40 years of age in 2015 and with no prior history of major cardiovascular diseases (ischaemic heart disease or stroke). The NHMS 2015 did not require the participants to state-specific types of heart attack.

We excluded data from 10,142 male and 11,065 female participants if they were younger than 40 years or if their data had missing values for any of predictors in the Globorisk risk prediction model (age, systolic blood pressure, total cholesterol level, history of diabetes mellitus, and smoking status). Ultimately, 4083 men and 4170 women were eligible for inclusion in the study because they met the Globorisk risk prediction model (Figure 1).

### 2.3. The Globorisk Risk Prediction Model to Calculate 10-Year CVD Risk

The Globorisk risk prediction model is based on a review of CVD risk variables at the outset. It calculates a population’s risk of cardiovascular disease in a given year (linked to each year in which the data were collected). Each risk factor’s proportional impact on the risk of cardiovascular disease is quantified by the model’s coefficients, which are the hazard ratios (unique to a population). The estimated coefficients are available on pages 4 and 5 in the paper [16].

To estimate the coefficients for the baseline CVD risk factors in the prediction equation, the Globorisk team pooled individual-level data from eight prospective cohorts—the Atherosclerosis Risk in Communities, Cardiovascular Health Study, Framingham Heart Study original cohort, Framingham Heart Study offspring cohort, Honolulu Heart Program, Multiple Risk Factor Intervention Trial, Puerto Rico Heart Health Program, and Women’s Health Initiative Clinical Trial [16,17]. All eight cohorts were conducted in the US mainland and Hawaii (the Honolulu Heart Program). The reference population for each cohort is different, but generally, American adults.

For the Globorisk risk prediction to calculate country-specific fatal and non-fatal CVD, we used the Malaysia average CVD rate in each age-and-sex group for 2015 provided by the Globorisk team. The method to estimate the average CVD rate is available in the Appendix A document from the Globorisk team [17]. After we replace the values of sex, age, smoking status, systolic blood pressure, diabetes status, and total cholesterol level in the Globorisk risk prediction for each eligible participant from the NHMS 2015 dataset, the prediction returns three primary outcomes: (1) the laboratory-based mean 10-year risk of fatal CVD, (2) the laboratory-based mean 10-year risk of fatal plus non-fatal CVD; and (3) the office-based mean 10-year risk of fatal plus non-fatal CVD. The fatal plus non-fatal Globorisk risk scores are the probability of future fatal or non-fatal CVD events relative to the fatal CVD events in the initial recalibration process for each country, respectively.

#### 2.3.1. The Laboratory-Based 10-Year Risk of Fatal CVD

This score is the 10-year risk for fatal cardiovascular disease only. Although fatal and non-fatal cardiovascular diseases are important for clinical and public health applications, national data for average death rates are more reliable than those for disease incidence, even in high-income countries [14,16]. The laboratory-based 10-year risk of fatal and non-fatal CVD Globorisk risk score is calculated from six variables: sex, age, smoking status, systolic blood pressure, diabetes status, and total cholesterol level.

#### 2.3.2. The Laboratory-Based 10-Year Risk of Fatal plus Non-Fatal CVD

The laboratory-based 10-year risk of fatal and non-fatal CVD Globorisk risk score is calculated from the same six variables: sex, age, smoking status, systolic blood pressure, diabetes status, and total cholesterol level. This calculation allows for the estimation of specific CVD risk using readily available population-wide survey data in most middle- and high-income countries.

#### 2.3.3. The Office-Based 10-Year Risk of Fatal plus Non-Non-Fatal CVD

The office-based 10-year risk of fatal and non-non-fatal CVD Globorisk risk score is calculated from five variables: sex, age, smoking status, systolic blood pressure, and BMI (diabetes status and total cholesterol are replaced with BMI) [22]. BMI has a strong association with diabetes status and total cholesterol and acts as a proxy for increased body weight, blood glucose, and serum cholesterol [17,22]. The modification estimates the CVD risk score in an economically poor resource setting in which laboratory facilities are limited.

### 2.4. Statistical Methods

Categorical variables are presented using frequencies (n) and percentages (%). Meanwhile, continuous variables are presented using means (SD) for normally distributed data and medians (interquartile range) for skewed data. We calculated the CVD risk scores for each eligible participant. The CVD risk scores are typically classified into various categories. For this paper, we divided them into four categories: a low risk for future CVD (if the CVD risk score was <10%) and 3 high-risk CVD categories (if CVD risk was equal to or greater than 10%, 20%, and 30%) [16,17,23]. An analysis was performed for overall risk and by sex, ethnicity, region, and state in Malaysia. All the statistical analyses were performed using R software version 3.6.1 and the gtsummary, summarytools, and ggplot2 packages in RStudio IDE [24,25,26,27].

### 2.5. Ethical Approval

This research was carried out in conformity with the Helsinki Declaration standards. The Human Research and Ethics Committee of Universiti Sains Malaysia USM/JEPeM/19100607 and the Medical Research and Ethics Committee of the National Institute of Health, Ministry of Health Malaysia NMRR-19-3061-51277 provided ethical approval (IIR).

## 3. Results

### 3.1. The Participants’ Characteristics

We performed CVD risk prediction on 8253 individuals aged between 40 and 84 years. The mean age for men was 53.72 years (SD = 9.32) and for women 52.06 years (SD = 8.32). Ethnically, the sample was predominated by Malay (62.3%), followed by Chinese (18.2%), Indian (7.2%), and other ethnicities.

The geographical areas were states, the principal administrative divisions of the country, of which there are 13, and 3 federal territories (WP Kuala Lumpur, WP Putrajaya and Labuan). The proportion of the study population in the state of Selangor was the highest (12.0%), while Labuan had the lowest (0.2%), which was proportionate to their overall population size. The distribution of people by ethnicity between men and women was fairly equal (Table 1).

In terms of cardiovascular risk factors, men had a higher prevalence of smoking (42%) than women (1%). The prevalence of diabetes was equivalent between the sexes: 28% for men and 29% for women. The overall mean for systolic blood pressure was 135.65 mm Hg (SD = 23.74), for total cholesterol 5.41 mmol (SD = 2.82). and BMI 27.33 kg/m^2^ (SD = 5.23). These values were higher in women compared with men (Table 1).

### 3.2. The 10-Year CVD Risk Prediction Score at the National Level

Table 2 shows that the average risk laboratory-based mean (SD) 10-year risk of fatal CVD, laboratory-based mean 10-year risk of fatal plus non-fatal CVD, and office-based mean 10-year risk of fatal plus non-fatal CVD were 0.07 (SD = 0.10), 0.14 (SD = 0.12), and 0.11 (SD = 0.09), respectively. The CVD risk for women appears to be lower than that of the men: mean 10-year risk of fatal CVD = 0.05 (SD = 0.09), laboratory-based mean 10-year risk of fatal plus non-fatal CVD = 0.11 (SD = 0.12), and office-based mean 10-year risk of fatal plus non-fatal CVD = 0.08 (SD = 0.09). The CVD risk categories (calculated from Globorisk risk prediction) showed that more men had high CVD risk scores for all Globorisk predicted risk. For example, their unified high-risk laboratory-based 10-year risk of fatal CVD was 29.3%, the unified high-risk laboratory-based 10-year risk of fatal plus non-fatal CVD was 73.0%, and the unified high-risk office-based ten-year risk of fatal plus non-fatal CVD was 72.0%.

Table 3 depicts the CVD risk for different ethnicities in Malaysia. The Malays had the highest Globorisk risk prediction compared with their counterparts with a mean of 10-year risk of fatal CVD = 0.07 (SD = 0.10), mean of laboratory-based 10-year risk of fatal plus non-fatal CVD = 0.15 (SD = 0.13), and mean of office-based 10-year risk of fatal plus non-fatal CVD = 0.12 (SD = 0.09). Malays also had the biggest percentage of high CVD risk scores in all Globorisk outcomes when compared with other ethnicities. Their unified high-risk laboratory-based 10-year risk of fatal CVD was 25.3%, unified high-risk laboratory-based 10-year risk of fatal plus non-fatal CVD 58.0%, and unified high-risk office-based 10-year risk of fatal plus non-fatal CVD 50.8%.

In Table 4, we present the region-specific CVD risk based on the Globorisk risk prediction. It shows that the northern region of Malaysia had the highest CVD risk scores compared with other regions in terms of mean (10-year risk of fatal Globorisk = 0.08 (SD = 0.11), laboratory-based 10-year risk of fatal plus non-fatal CVD = 0.15 (SD = 0.13), and office-based 10-year risk of fatal plus non-fatal CVD = 0.12 (SD = 0.10)) and category (unified high-risk laboratory-based 10-year fatal Globorisk = 26.9%, unified high-risk laboratory-based 10-year risk for CVD = 57.7%, and unified high-risk office-based 10-year risk for fatal plus non-fatal CVD = 53.2%). We created a dashboard that allows users to view the results of this study in an inter-active way (Appendix A). Due to space constraints, the dashboard also includes results that are not available in this publication.

The states in the northern region (Perlis, Kedah, Penang, and Perak) showed the highest proportion of high CVD risk for all Globorisk scores compared with the other states. All related details regarding state-specific Globorisk risk predictions are available in Table 5.

## 4. Discussion

From our analysis, the population-based data for 8253 adults across Malaysia showed variations in the estimated 10-year CVD risk scores. The overall CVD risk for the Malaysian population is higher than in Japan, South Korea, Spain, and Denmark [16,17]. Common comorbidities in the form of diabetes, hypertension, hypercholesterolemia, obesity, and smoking are prevalent in Malaysia [28]. In fact, a substantial increase in those common CVD risk factors was observed in the data from the NHMS conducted among Malaysians [29,30]. The increase in these CVD risk factors could be attributed to the sedentary lifestyle common among Malaysians [31,32,33]. It is timely that the CVD control and prevention programmes in Malaysia consider a newer and more effective approach, such as the multifactorial intensive therapy pre-specified algorithm, for the management of CVD risk factors, including non-pharmacological and pharmacological treatment [34].

Men have higher CVD risk scores than women in Malaysia. This result is in line with previous studies [3,6,35]. For the 10-year risk of fatal CVD, our findings for men and women with a high-risk CVD risk score (See Table 2) were lower than those of South Korea (men: 7.0%, women: 7.0%) and China (men: 33.0%, women: 28.0%) [16]. As for the laboratory-based 10-year risk of fatal and non-fatal CVD, our high-risk CVD scores among men and women were higher than those in South Korea (men: 0.3%, women: 0.5%) and China (men: 10.3%, women: 9.2%) [17].

The Malaysian sex-specific high risk for the office-based 10-year risk of fatal plus non-fatal CVD was higher than in South Korea (men: 0.1%, women: 0.1%) but was substantially lower than in China (men: 8.8%, women: 6.5%) [17]. In general, in all comparisons, the men had higher high-risk CVD risk scores than women. In this study, it was specifically because men had higher baseline risk factors of CVD compared with women, particularly regarding the smoking rate. In addition, the constellation of smoking with other risk factors increases the CVD risk score and the risk of CVD events in the foreseeable future [32,36,37]. In particular, smoking is associated with increases in oxidative stress, thus predisposing individuals who smoke to developing cardiovascular diseases [38,39].

Meanwhile, the comparison between ethnicities in Malaysia showed that Malays have the highest CVD risk of all the CVD risk groups (Table 3). This finding was consistent with previous studies showing that the high 10-year CVD risk among Malays was due to higher baseline CVD risk factors, such as diabetes, hypertension, hypercholesterolemia, and smoking [3,4,8]. The prominent CVD risk factors among Malays were due to a high rate of unawareness of having NCDs and poor health-seeking behaviour [40,41,42]. Eating foods high in saturated fat, trans fat, salt, and sugar is also associated with a high risk of CVD. Previous studies conducted among Malaysians showed that Malays had poor eating habits associated with the risk of developing CVD [43,44].

Regarding the region-specific analysis, the northern region, which comprises the states of Penang, Kedah, Perlis, and Perak, has the highest CVD risk scores compared with its counterparts (Table 4 and Table 5). The highest proportion of individuals with high CVD risk in the northern region is due to a high proportion of four common CVD risk factors: diabetes, hypertension, hypercholesterolemia, and smoking, as evidenced in the NHMS 2015 and its corresponding cross-sectional study [30,45].

Estimating 10-year fatal and fatal plus non-fatal laboratory and office CVD risk scores reveals a comparable estimate for low and high scores. This finding is in line with previous research that showed that 80% of adults were comparably classified into low and high CVD risk by laboratory and office risk scores [46]. Thus, the office Globorisk risk score allows for risk prediction in settings where there is no access to laboratory testing, such as during community screening and home care visits, which subsequently reduces the cost of laboratory testing.

Our study has several strengths and limitations. To the best of our knowledge, this study is the first of its kind in Malaysia that used the Globorisk risk prediction model to estimate CVD risk. Our study also used large data from the 2015 NHMS, which is representative of the Malaysian adult population. The application of the Globorisk risk prediction model used in our paper is bounded to the limitation of estimation of fatal and non-fatal CVD rates using national ischaemic heart disease and stroke death rates from the WHO. The Globorisk risk score was developed and validated using data from eight large prospective cohort studies, but none was from the Southeast Asia region, hence reducing its external validity. The Globorisk risk prediction also predicts the 10-year CVD risk; however, 10-year risks underestimate lifetime risk and might, therefore, lead to undertreatment, especially in younger individuals. Future studies should include the validation of Globorisk risk scores in the Southeast Asian population and the application of a risk score to the previous NHMS data to enable trend analysis and the translation of risk into the actual burden of CVD in Malaysia.

## 5. Conclusions

The 10-year risk for fatal and fatal plus non-fatal CVD based on the Globorisk risk prediction model shows substantial differences in the average CVD risk and CVD risk categories for sex, ethnicity, region, and state. Malaysian men, Malays, and those living in the northern region have a higher CVD risk than their counterparts. The results of this study are representative of the Malaysian adult population and would be useful for the control and prevention of CVD in Malaysia.

## Figures and Tables

**Figure 1 medicina-58-00656-f001:**
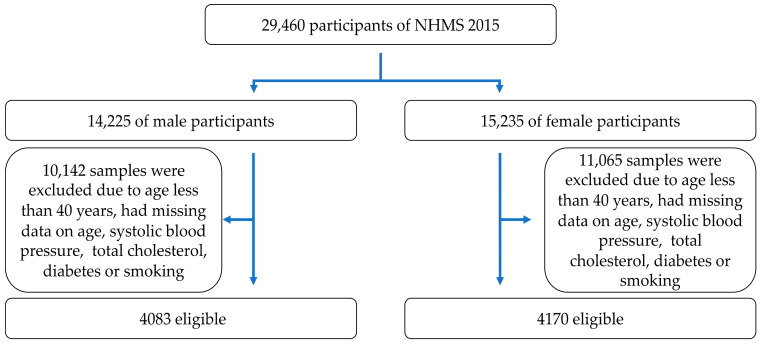
Study participant flow chart.

**Table 1 medicina-58-00656-t001:** The participants’ characteristics.

Characteristic	Overall (*n* = 8253)	Women (*n* = 4170)	Men (*n* = 4083)
Age, mean (SD), years	52.88 (8.87)	52.06 (8.32)	53.72 (9.32)
Ethnic			
Chinese	1496 (18.2%)	736 (17.6%)	760 (19.0%)
Indian	591 (7.2%)	320 (7.7%)	271 (6.6%)
Malay	5149 (62.3%)	2608 (62.6%)	2541 (61.8%)
Others ^a^	357 (4.3%)	167 (4.0%)	190 (4.7%)
Others Bumis ^b^	660 (8.0%)	339 (8.1%)	321 (7.9%)
State			
Johor	751 (9.1%)	382 (9.2%)	369 (9.0%)
Kedah	561 (6.8%)	285 (6.8%)	276 (6.8%)
Kelantan	516 (6.3%)	258 (6.2%)	258 (6.3%)
Labuan	15 (0.2%)	6 (0.1%)	9 (0.2%)
Melaka	447 (5.4%)	252 (6.0%)	195 (4.8%)
Negeri Sembilan	568 (6.9%)	318 (7.6%)	250 (6.1%)
Pahang	497 (6.0%)	227 (5.4%)	270 (6.6%)
Penang	629 (7.6%)	332 (8.0%)	297 (7.3%)
Perak	650 (7.9%)	327 (7.8%)	323 (7.9%)
Perlis	624 (7.6%)	310 (7.4%)	314 (7.7%)
Sabah	593 (7.2%)	297 (7.1%)	296 (7.2%)
Sarawak	506 (6.1%)	252 (6.0%)	254 (6.2%)
Selangor	970 (12.0%)	468 (11.0%)	502 (12.0%)
Terengganu	526 (6.4%)	257 (6.2%)	269 (6.6%)
WP Kuala Lumpur	278 (3.4%)	145 (3.5%)	133 (3.3%)
WP Putrajaya	122 (1.5%)	54 (1.3%)	68 (1.7%)
Smoking Status, yes	1741 (21.0%)	41 (1.0%)	1700 (42.0%)
Diabetes Status, yes	2350 (28.0%)	1219 (29.0%)	1131 (28.0%)
Systolic Blood Pressure, mean (SD), mm Hg	135.31 (23.17)	135.65 (23.74)	134.97 (22.58)
Total Cholesterol, mean (SD), mmol/L	5.18 (2.86)	5.41 (2.82)	4.94 (2.89)
BMI, mean (SD), kg/m^2^	26.61 (4.94)	27.33 (5.23)	25.87 (4.51)

^a^ Other ethnicities, comprising other Malaysian minorities such as Sikh, Baba, Chitty, Eurasian, and non-citizens. ^b^ Other Bumiputera, comprising more than 40 indigenous ethnicities that reside in both Peninsular and Borneo, Malaysia.

**Table 2 medicina-58-00656-t002:** The overall and sex-specific 10-year CVD risk.

Prediction	Overall (*n* = 8253)	Female (*n* = 4170)	Male (*n* = 4083)
10-year fatal (lab-based) CVD risk, Mean (SD)	0.07 (0.10)	0.05 (0.09)	0.08 (0.10)
Categories			
Low Risk < 10%	6392 (77.0%)	3515 (84.4%)	2877 (70.7%)
High Risk A ≥ 10%	1123 (14.0%)	381 (9.1%)	742 (18.0%)
High Risk B ≥ 20%	411 (5.0%)	148 (3.5%)	263 (6.4%)
High Risk C ≥ 30%	327 (4.0%)	126 (3.0%)	201 (4.9%)
10-year fatal plus non-fatal (lab-based) CVD risk, Mean (SD)	0.14 (0.12)	0.11 (0.12)	0.17 (0.12)
Categories			
Low Risk < 10%	3732 (45.0%)	2617 (63%)	1115 (27.0%)
High Risk A ≥ 10%	2502 (30.0%)	884 (21%)	1618 (40.0%)
High Risk B ≥ 20%	1117 (14.0%)	329 (7.9%)	788 (19.0%)
High Risk C ≥ 30%	902 (11.0%)	340 (8.1%)	562 (14.0%)
10-year fatal plus non-fatal (office-based) CVD risk, Mean (SD)	0.11 (0.09)	0.08 (0.09)	0.14 (0.08)
Categories			
Low Risk < 10%	4205 (51.1%)	3072 (74.0%)	1133 (28.0%)
High Risk A ≥ 10%	2724 (33.0%)	764 (18.0%)	1960 (48.0%)
High Risk B ≥ 20%	920 (11.0%)	187 (4.5%)	733 (17.7%)
High Risk C ≥ 30%	404 (4.9%)	147 (3.5%)	257 (6.3%)

**Table 3 medicina-58-00656-t003:** The ethnicity-specific 10-year CVD risk.

Prediction	Chinese (*n* = 1496)	Indian (*n* = 591)	Malay (*n* = 5149)	Others ^a^ (*n* = 357)	Others Bumis ^b^ (*n* = 660)
10-year fatal (lab-based) CVD risk, Mean (SD)	0.06 (0.09)	0.06 (0.09)	0.07 (0.10)	0.04 (0.06)	0.05 (0.07)
Categories					
Low Risk < 10%	1214 (81.0%)	469 (79.0%)	3846 (74.7%)	310 (86.9%)	553 (83.7%)
High Risk A ≥ 10%	166 (11.3%)	71 (12.4%)	773 (15.0%)	36 (10.0%)	77 (11.8%)
High Risk B ≥ 20%	60 (4.0%)	25 (4.2%)	299 (5.8%)	5 (1.4%)	22 (3.3%)
High Risk C ≥ 30%	56 (3.7%)	26 (4.4%)	231 (4.5%)	6 (1.7%)	8 (1.2%)
10-year fatal plus non-fatal (lab-based) CVD risk, Mean (SD)	0.12 (0.11)	0.13 (0.12)	0.15 (0.13)	0.11 (0.10)	0.11 (0.10)
Categories					
Low Risk < 10%	760 (50.8%)	275 (47.0%)	2161 (42.0%)	196 (54.7%)	340 (51.4%)
High Risk A ≥ 10%	446 (29.8%)	174 (29.0%)	1573 (31.0%)	99 (27.6%)	210 (31.8%)
High Risk B ≥ 20%	168 (11.2%)	79 (13.0%)	756 (15.0%)	42 (12.1%)	72 (11.0%)
High Risk C ≥ 30%	122 (8.2%)	63 (11.0%)	659 (13.0%)	20 (5.6%)	38 (5.8%)
10-year fatal plus non-fatal (office-based) CVD risk, Mean (SD)	0.11 (0.09)	0.10 (0.09)	0.12 (0.09)	0.09 (0.07)	0.10 (0.08)
Categories					
Low Risk < 10%	779 (52.3%)	335 (56.8%)	2512 (49.2%)	222 (62.2%)	357 (54.3%)
High Risk A ≥ 10%	500 (33.3%)	176 (29.7%)	1723 (33.4%)	111 (31.1%)	214 (32.3%)
High Risk B ≥ 20%	156 (10.3%)	48 (8.1%)	642 (12.1%)	15 (4.2%)	59 (8.9%)
High Risk C ≥ 30%	61 (4.1%)	32 (5.4%)	272 (5.3%)	9 (2.5%)	30 (4.5%)

^a^ Other ethnicities, comprising other Malaysian minorities such as Sikh, Baba, Chitty, Eurasian, and non-citizens. ^b^ Other Bumiputera, comprising more than 40 indigenous ethnicities that reside in both Peninsular and Borneo, Malaysia.

**Table 4 medicina-58-00656-t004:** The region-specific 10-year CVD risk.

Prediction	Centre (*n* = 1370)	East Coast (*n* = 1539)	Northern (*n* = 2464)	Sabah (*n* = 608)	Sarawak (*n* = 506)	Southern (*n* = 1766)
10-year fatal (lab-based) CVD risk, Mean (SD)	0.05 (0.08)	0.07 (0.09)	0.08 (0.11)	0.05 (0.08)	0.06 (0.08)	0.07 (0.10)
Categories						
Low Risk < 10%	1131 (82.9%)	1195 (77.5%)	1800 (73.1%)	503 (82.8%)	418 (82.5%)	1345 (76.0%)
High Risk A ≥ 10%	157 (11.1%)	208 (13.7%)	372 (15.0%)	73 (11.9%)	54 (10.8%)	259 (14.8%)
High Risk B ≥ 20%	37 (2.7%)	74 (4.8%)	170 (6.9%)	21 (3.5%)	23 (4.5%)	86 (4.9%)
High Risk C ≥ 30%	45 (3.3%)	62 (4.0%)	122 (5.0%)	11 (1.8%)	11 (2.2%)	76 (4.3%)
10-year fatal plus non-fatal (lab-based) CVD risk, Mean (SD)	0.12 (0.11)	0.14 (0.12)	0.15 (0.13)	0.11 (0.10)	0.12 (0.11)	0.14 (0.13)
Categories						
Low Risk < 10%	712 (52.0%)	655 (42.8%)	1028 (42.3%)	311 (50.8%)	257 (51.1%)	769 (43.6%)
High Risk A ≥ 10%	383 (27.8%)	517 (33.8%)	718 (28.8%)	187 (30.6%)	153 (30.0%)	544 (30.7%)
High Risk B ≥ 20%	163 (12.0%)	208 (13.6%)	372 (14.7%)	70 (12.0%)	57 (11.2%)	247 (13.6%)
High Risk C ≥ 30%	112 (8.2%)	159 (9.8%)	346 (14.2%)	40 (6.6%)	39 (7.7%)	206 (12.1%)
10-year fatal plus non-fatal (office-based) CVD risk, Mean (SD)	0.10 (0.08)	0.11 (0.09)	0.12 (0.10)	0.10 (0.08)	0.10 (0.08)	0.11 (0.09)
Categories						
Low Risk < 10%	774 (56.3%)	780 (51.3%)	1168 (46.8%)	329 (54.2%)	265 (52.0%)	889 (50.3%)
High Risk A ≥ 10%	441 (32.4%)	515 (33.2%)	801 (32.7%)	196 (32.2%)	171 (34.2%)	600 (34.1%)
High Risk B ≥ 20%	108 (7.9%)	176 (11.1%)	336 (14.0%)	55 (9.0%)	47 (9.3%)	198 (11.1%)
High Risk C ≥ 30%	47 (3.4%)	68 (4.4%)	159 (6.5%)	28 (4.6%)	23 (4.5%)	79 (4.5%)

**Table 5 medicina-58-00656-t005:** The state-specific 10-year CVD risk.

States	*n*	10-Year Fatal (Lab-Based) CVD Risk		10-Year Fatal and Non-Fatal (Lab-Based) CVD Risk		10-Year Fatal and Non-Fatal (Office-Based) CVD Risk	
		Low Risk	High Risk	Low Risk	High Risk	Low Risk	High Risk
Johor	729	706 (97.0%)	23 (3.0%)	652 (89.0%)	77 (11.0%)	700 (96.0%)	29 (4.0%)
Kedah	521	492 (94.4%)	29 (5.6%)	435 (82.9%)	86 (17.1%)	492 (94.4%)	29 (5.6%)
Kelantan	513	484 (94.3%)	29 (5.7%)	451 (87.8%)	62 (12.2%)	489 (95.3%)	24 (4.7%)
Labuan	15	15 (100%)	0 (0%)	15 (100%)	0 (0%)	15 (100%)	0 (0%)
Melaka	416	397 (95.4%)	19 (4.6%)	370 (89.0%)	46 (11.0%)	400 (96.2%)	16 (3.8%)
Negeri Sembilan	544	513 (94.3%)	31 (5.7%)	472 (86.7%)	72 (13.3%)	515 (94.7%)	29 (5.3%)
Pahang	482	466 (97.0%)	16 (3.0%)	441 (91.5%)	41 (8.5%)	463 (96.0%)	19 (4.0%)
Penang	602	573 (95.2%)	29 (4.8%)	538 (89.0%)	64 (11.0%)	565 (93.8%)	37 (6.2%)
Perak	636	608 (96.0%)	28 (4.0%)	559 (87.6%)	77 (12.4%)	593 (93.2%)	43 (6.8%)
Perlis	618	590 (95.5%)	28 (4.5%)	535 (87.3%)	83 (12.7%)	588 (95.0%)	30 (5.0%)
Sabah	571	561 (98.2%)	10 (1.8%)	535 (93.5%)	36 (6.5%)	546 (96.1%)	25 (3.9%)
Sarawak	501	490 (98.0%)	11 (2.0%)	465 (93.0%)	36 (7.0%)	482 (96.2%)	19 (3.8%)
Selangor	922	886 (96.0%)	36 (4.0%)	844 (91.5%)	78 (8.5%)	891 (96.6%)	31 (3.4%)
Terengganu	515	501 (97.3%)	14 (2.7%)	472 (91.7%)	43 (8.3%)	502 (97.0%)	13 (3.0%)
WP Kuala Lumpur	270	264 (98.0%)	6 (2.0%)	250 (93.0%)	20 (7.0%)	264 (98.0%)	6 (2.0%)
WP Putrajaya	122	122 (100%)	0 (0%)	114 (93.4%)	8 (6.6%)	121 (99.2%)	1 (0.8%)

## Data Availability

All the data are available from the corresponding author upon reasonable request.

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
