# Peer review of "The Ten-Year Risk Prediction for Cardiovascular Disease for Malaysian Adults Using the Laboratory-Based and Office-Based (Globorisk) Prediction Model"

_medicina, 2022, doi:10.3390/medicina58050656_

Round 1

Reviewer 1 Report

The authors have adequately addressed my comments and I am pleased to offer only minor comments for consideration. The paper is well-written and the Discussion is clear and succinct.

Comment 1: [Abstract, line 14 and Introduction, line 49] I am not sure how ‘novel’ or ‘unique’ the Globorisk is, as the authors themselves are well-aware that many other cardiovascular disease (CVD) risk predictions are abound. In my opinion, the most compelling reason to use Globorisk is that it has been validated in many populations, but even this is not a novelty or uniqueness in itself as other risk functions, like Framingham, is similarly well validated. I would suggest the authors revise this and state that the reason for choosing Globorisk is how well validated it is.

Comment 2: [Materials and Methods, lines 82 to 90] Can the authors briefly comment on using finger-prick blood for testing blood glucose and blood lipids as opposed to venous blood?

Comment 3: [Materials and Methods, lines 97 to 101] Since the office-based score does not require data on diabetes status and blood lipid levels, wouldn’t the authors have unnecessarily excluded participants for missing diabetes status and blood lipids for the office-based score?

Comment 4: [Materials and Methods, lines 110 to 111] Can the authors briefly comment on how the populations of the eight countries may be similar or dissimilar to Malaysia’s?

Author Response

First and foremost, thank you for taking the time to read through our work and offer suggestions for improvement.
The enclosed document contained our point-by-point responses to the comments.
Please see the attachment for further information.

Reviewer 2 Report

No further comments.

Author Response

Thank you for taking the time to read through the manuscript. We appreciate the suggestions for publishing our research in medicina.

This manuscript is a resubmission of an earlier submission. The following is a list of the peer review reports and author responses from that submission.

Round 1

Reviewer 1 Report

In this study, the authors evaluated laboratory-based and office-based risk scores to predict the 10-year risk of fatal cardiovascular disease (CVD) and fatal plus non-fatal CVD in the Malaysian adult population using Globorisk, a risk prediction score aimed at calculating CVD risk. Data was from the National Health and Morbidity Survey 2015 (NHMS 2015), a nationwide population-based cross-sectional survey performed in 2015. In general, the analysis is straightforward and appropriate. However, I am not fully convinced by certain arguments that the authors have made. These concerns are detailed below.

Comment 1: [Introduction, lines 39 to 40] Can the authors briefly discuss their rationale of using Globorisk over other similarly well-validated CVD risk predictors, such as the Pooled Cohort equations?

Comment 2: [Introduction, lines 40 to 43] I agree with these statements that the authors have made, that risk prediction models work best in the populations they were derived in, and perform poorly in other populations perhaps due to differences in population characteristics, differences in associations with the risk factors included in the risk prediction models, and differences in baseline risk. There is therefore often a need to validate and recalibrate risk prediction models to ensure good fit [1]. However, when I checked Globorisk’s website, Globorisk does not appear to have been validated in Malaysia. Hence, I am not convinced that using Globorisk solves these issues the authors have brought up.

Comment 3: [Introduction, lines 47 to 49] I do not understand what the authors mean. I know that this was taken from Globorisk’s website, but I do not think Globorisk is “the first and only unique cardiovascular disease risk score that predicts the risk of heart attack or stroke in healthy people (those who haven’t experienced a heart attack or stroke) in all countries throughout the world.”

Comment 4: [Introduction, line 52] “Assume the person hasn’t had a recent diabetes or cholesterol test.” is not a complete sentence.

Comment 5: [Materials and Methods, line 90] What is the rationale for excluding participants less than 40 years of age?

Comment 6: [Materials and Methods, line 91] Does this exclusion criteria include acute myocardial infarction (AMI)?

Comment 7: [Materials and Methods, line 104 to 105] Where were the beta-coefficients obtained, if Globorisk has not been validated in Malaysia?

Comment 8: [Materials and Methods, line 107 to 112] Since the Globorisk is based on various prospective cohorts and the authors are instead using cross-sectional data, can the authors comment on how appropriate this method may be?

Comment 9: [Results, line 171] Can the authors correct the error, “Error! Reference source not found”? More instances of this error can be found in the rest of this section.

Comment 10: [Discussion, general] In general, the Discussion is well-written. To emphasize on the clinical implications of the paper, have the authors considered translating these risk scores to actual incidences? Have the authors also considered using the NHMS data from other years, and checked if the risk scores are generally consistent? It would be interesting even if risk scores changed, as it may signify change in lifestyle risk factors.

  1. Kengne, A.P., et al., Non-invasive risk scores for prediction of type 2 diabetes (EPIC-InterAct): a validation of existing models. Lancet Diabetes Endocrinol, 2014. 2(1): p. 19-29.

Reviewer 2 Report

Based on a nationwide population-based cross-sectional survey in Malaysia, the authors estimated 10-years CV risk in subjects older than 40 and with no history of CVD. They assessed a very heterogeneous 10-year CV risk among the various populations and geographic areas studied. These results are a call to action towards uniform control of CV risk factors. The conclusions are supported by results. Figure and tables are clear.

This review raises only a few issues that need to be addressed.

1- Personalized therapy represents an upgrade as well as a challenge in the management of people at CV risk. Very recently, in NID-2 study (Cardiovasc Diabetol (2021) 20: 145. doi: 10.1186 / s12933-021-01343 -1) mortality and MACEs were assessed in a population in primary CV prevention, but at very high CV risk. This randomized multicenter study in which the two genders were equally represented originally demonstrated the ability of a multifactorial therapeutic approach to improve MACEs and overall mortality with short intervention, as well as a long duration of CV protection. Therefore, comprehensive therapy aimed at targeting all major CV risk factors in order to improve mortality and MACEs should be mandatory. This important issue and the above reference need to be adequately commented on in the discussion and conclusions.

2- Lines 171, 177, 180, 194 and 206. Please, add references.

3- A linguistic revision by a native English speaker is required.